# Short Working Memory Impairment Associated with Hippocampal Microglia Activation in Chronic Hepatic Encephalopathy

**DOI:** 10.3390/metabo14040193

**Published:** 2024-03-29

**Authors:** Bilal El-Mansoury, Kamal Smimih, Abdelaati El Khiat, Ahmed Draoui, Abdelmohcine Aimrane, Redouane Chatoui, Abdesslam Ferssiwi, Abdelali Bitar, Halima Gamrani, Arumugam R. Jayakumar, Omar El Hiba

**Affiliations:** 1Laboratory of Anthropogenic, Biotechnology and Health, Nutritional Physiopathologies, Neurosciences and Toxicology Team, Faculty of Sciences, Chouaib Doukkali University, Av. Des Facultés, El Jadida 24000, Morocco; el-mansouri.b@ucd.ac.ma (B.E.-M.); a.elkhiat.ced@uca.ac.ma (A.E.K.); aimrane.a@ucd.ac.ma (A.A.); ferssiwi.a@ucd.ac.ma (A.F.); bitar.a@ucd.ac.ma (A.B.); 2Laboratory of Genie-Biology, Faculty of Sciences and Techniques, Sultan Moulay Slimane University, Beni Mellal 23000, Morocco; kamal.smimih@usms.ma (K.S.); redouane.chatoui@usms.ac.ma (R.C.); 3Higher Institute of Nursing Professions and Health Techniques, Ministry of Health, Ouarzazate 45000, Morocco; 4Laboratory of Clinical and Experimental Neurosciences and Environment, Faculty of Medicine and Pharmacy, Cadi Ayyad University, Marrakech 40000, Morocco; 5Laboratory of Clinical and Experimental Neurosciences and Environment, Faculty of Science Semlalia, Cadi Ayyad University, Marrakech 40000, Morocco; a.draoui.ced@uca.ac.ma (A.D.); gamrani@uca.ac.ma (H.G.); 6Miller School of Medicine, University of Miami, Miami, FL 33136, USA

**Keywords:** hepatic encephalopathy, biochemical analysis, cirrhosis, memory assessment, short working memory, microglia activation, Iba1

## Abstract

Hepatic encephalopathy (HE) is a major neuropsychological condition that occursas a result of impaired liver function. It is frequently observed in patients with advanced liver disease or cirrhosis. Memory impairment is among the symptoms of HE; the pathophysiologic mechanism for this enervating condition remains unclear. However, it is possible that neuroinflammation may be involved, as recent studies have emphasized such phenomena. Therefore, the aim of the present study is to assess short working memory (SWM) and examine the involvement of microglia in a chronic model of HE. The study was carried out with male Wistar rats that were induced by repeated thioacetamide (TAA) administration (100 mg/kg i.p injection for 10 days). SWM function was assessed through Y-maze, T-Maze, and novel object recognition (NOR) tests, together with an immunofluorescence study of microglia activation within the hippocampal areas. Our data showed impaired SWM in TAA-treated rats that was associated with microglial activation in the three hippocampal regions, and which contributed to cognitive impairment.

## 1. Introduction

Hepatic encephalopathy (HE) is a devastating neurological complication of liver dysfunction that presents in acute and chronic forms [1,2]. Type C HE develops slowly under cirrhosis and includes a wide range of cognitive manifestations [3] such as short working memory (SWM) deterioration [4,5]. Memory disturbances were detected in patients with HE as well as in several animal models of this disorder. However, the exact mechanism underpinning memory impairment in HE is poorly understood.

Several etiopathologic factors are believed to contribute to the onset and the development of HE. An elevated brain ammonia level constitutes the main and less controverted etiological factor in the pathogenesis of HE. However, a number of studies emphasized the significant involvement of neuroinflammation in the development of HE [6,7,8].

In the central nervous system (CNS), the inflammatory process is mediated mainly by microglial cells [9]. Microglia are the primary innate immune cells in the CNS [7,10], and are involved in the development of neuroinflammation in HE. Microglia activation is known as the main source of proinflammatory cytokines, which contribute to the development of neuroinflammation [11,12]. It is worth mentioning that a growing body of evidence supports the involvement of microglia activation and subsequent neuroinflammation in the pathogenesis of HE. Indeed, recent studies have shown that neuroinflammation plays a significant role in the pathogenesis of both acute and chronic HE [7,13,14,15,16,17,18]. Microglia can be activated by several factors, including pro-inflammatory cytokines (derived as a consequence of liver necrosis and/or sepsis), lipopolysaccharide (LPS; endotoxin), and ammonia [7].

It is well-known that microglial cells are crucial for cerebral plasticity and learning [8]; hence, microglia dysfunction can lead to cognitive abnormalities. It is believed that chronic liver disease (CLD) could trigger both systemic and CNS proinflammatory processes, contributing to cognitive impairment in HE [19]. More importantly, microglia activation and neuroinflammation are thought to impair SWM [20,21]. However, nothing is currently known about the involvement of neuroinflammation in the development of memory impairment in type C HE. 

Since the hippocampus is the critical brain area for learning and memory and, currently, there are no studies investigating the neuroinflammatory process within the hippocampus and its causative effect on short memory impairment in type C HE, we examined SWM in an animal model of type C HE and its association with the inflammation status (microglial activation). 

## 2. Materials and Methods

### 2.1. Chemicals

Thioacetamide (TAA) (Sigma, Darmstadt, Germany, Lot#68H3492) and Dextrose Anhydrous (LobaChemie, Mumbai, India, product code#0316801000).

### 2.2. Animals

Adultmale Wistar rats weighing 160–200 g were obtained from the central animal-care facilities of the Faculty of Science, Chouaib Doukkali University, El Jadida, Morocco. Rats were housed at a constant room temperature (25 °C) on a 12 h dark–light cycle with ad libitum access to water and food. Animals were treated in compliance with the Moroccan Research Ethics Committee of the Moroccan Society of Ethics and Animal Research (MoSEAR Ref: UCD-FS-05/2023, 10 January 2023).

### 2.3. Induction of Type C HE

Induction of chronic hepatic encephalopathy was performed using the hepatotoxin thioacetamide (TAA), as described in the International Society for Hepatic Encephalopathy and Nitrogen Metabolism (ISHEN) guidelines for experimental models of HE [22]. Rats were divided into 2 groups as follows: -Group 1 (TAA): Male Wistar rats (n = 8, 162.48 ± 11.60 g), received repeated intraperitoneal injections (i.p) of TAA solution (100 mg TAA/kg. prepared in 0.9% NaCl), separated by a 24 h interval for 10 consecutive days. In order to minimize weight loss, hypoglycemia, dehydration and renal failure, which may arise from liver failure, a solution of dextrose (5%), NaCl (0.45%) and KCl20 mEq/L) was added to drinking water for 10 days;-Group 2 (Control): Male Wistar rats (n = 8, 162.03 g ± 10.28 g) received an equal volume of 0.9% sodium chloride solution (NaCl 0.9%).

### 2.4. Assessment of Liver Function

#### 2.4.1. Biochemical Analysis

In order to assess liver function in the TAA-treated rats, animals were anesthetized by urethane (1 g/kg i.p) and blood samples were collected from the rats’ jugular veins. Blood samples were then centrifuged (3000× *g*, 4 °C, 20 min) to separate the serum. The sera were then subjected to biochemical markers, including: aspartate-amino-transferase (AST), alanine-amino-transferase (ALT), urea, and creatinine. Additionally, blood and brain ammonia were evaluated. In addition, we measured other serum and liver biomarkers in TAA-treated rats using ELISA, as described previously by [23], including: (apolipoprotein A1 (Apo A1), α-2-macroglobulin, hyaluronic acid, procollagen III amino terminal peptide, tissue inhibitor of metalloproteinase 1, matrix metalloproteinases 3 (MMP-3) and 9, chondrex (YKL40), connective tissue growth factor (CTGF), paraoxonase-1, multidrug resistance protein-2, and transforming growth factor beta-1. 

#### 2.4.2. Liver Morphometric Analysis

After deep anesthesia, the liver was gently dissected out and weighed. The liver-to-bodyweight ratio was calculated as [24]:LBWR = (liver weight/bodyweight) × 100

#### 2.4.3. Liver Histopathology

The livers of controls and TAA-treated rats were removed and fixed in 10% formalin, regularly subjected to paraffin sections, and stained with hematoxylin and eosin (H and E). Briefly, after gradient dehydration with various concentrations of alcohol in an automatic tissue dehydrator (HistoCore PELORIS 3 Premium Tissue Processing System, Leica Biosystems Inc., Buffalo Grove, IL, USA), the liver of each animal was embedded in paraffin blocks (using HistoCore Arcadia Embedding Center, Leica Biosystems Inc., Buffalo Grove, IL, USA). The liver tissue was then cut into 10 µm thin slices via an ultra-thin semiautomatic microtome (Histocoreautocut automated rotary microtome, Leica Biosystems Inc., Buffalo Grove, IL, USA), and adhered to the slides. Then, the slides were stained with H and E, and the morphological changes were evaluated using a microscope (Olympus VS120 Automated Slide Scanner, Olympus, Pittsburgh, PA, USA) [25,26].

### 2.5. Short Working Memory Assessment

#### 2.5.1. Novel Object Recognition Test

The novel object recognition test (NOR) is a well-established analysis for testing SWM in rodents. The test was performed according to the protocol described by Bevins and Besheer [27]. The open field (OF) apparatus (100 cm × 100 cm × 30 cm), and the objects used are made from heavy wood to prevent them from being moved by animals. In addition, two objects are similar and are different from the third one in color, shape, and appearance.

The test consists of three sessions: one habituation session, one training session, and one test session. For the habituation phase, each rat was placed individually in the OF without objects to explore it for 5 min. During the training session (for 10 min exploration), each rat was individually exposed to two identical objects placed at an equal distance in the OF apparatus. Concerning the test session, 1 h later, each animal was individually exposed to one of the two familiar objects (old objects) and the other one was replaced by a new object (novel), over the course of 5 min [27]. Normally, rodents have a spontaneous tendency to spend significantly more time exploring a novel object than a familiar one in the absence of externally applied rules or reinforcement. Hence, the choice of exploring the novel object reflects the use of learning and recognition memory.

The parameters that indicate object interaction or exploration (biting, sniffing, and touching the object with forepaws or nose) were noted. The average exploration time and the discrimination ratio were calculated for each group [27]. Memory is considered to be intact if there is no significant difference in the average interaction time between the new and the old objects [28]. NOR task performance was quantified using the discrimination ratio (DR), which reflects the duration of exploring the new object, as compared to the old one, in the total exploration time. 

#### 2.5.2. T-Maze Test

The T-maze test was also used to evaluate short working memory. It is well known that this test effectively detects short memory impairments in rodents [29,30]. The T-maze apparatus is made from wood with two lateral arms (50 cm long, 10 cm in width, with walls of 30 cm height) and one central arm [29]. This test is based on two tests given in rapid succession for each animal; in the second test, the rodent tends to choose a new arm rather than returning to the previously visited one, reflecting the memory of the first choice, referred to as spontaneous [24,31,32]. The percentage (%) of the spontaneous alternation between the left and right arms according to the arm visited during the previous session constitutes the analyzed parameter. These data were transformed into scores for further statistical analysis: a score of 0% corresponds to no alternation; a score of 100% corresponds to a correct alternation in each test. Control animals typically obtain around 80% correct responses [32]. 

#### 2.5.3. Y-Maze Test

To complement and reinforce the previous 2 tests, we also used the Y-maze test, which evaluated SWM by monitoring spontaneous alternation behavior. The apparatus is composed of three spaced black-painted wooden arms (identical arms) (120°) with a 50 cm length and 20 cm height. The arms of the maze are clearly designated as “A”, “B” and “C”. For the test session each animal is placed in arm “A” (start arm), with the snout facing the center for 10 min. The test assesses SWM by allowing rats to explore all three arms of the maze for 10 min. Normally, a rat with an intact memory will have the tendency to return directly to the arm previously visited, driven by their innate curiosity to explore a new environment (arm). Hence, over the course of multiple arm entries (an entry occurs when all the four limbs are within the arm), a normal rat should exhibit a tendency to enter a less recently visited arm. The number of arm entries and the number of correct alternations were recorded in order to calculate the percentage of alternations. The alternation behavior percentage (%)was calculated for each group [33,34].

### 2.6. Immunofluorescence

Neuroinflammation is known to impair memory function and may contribute to the development of HE. Accordingly, we tested for possible microglial changes in the hippocampus, the critical brain structure predominantly involved in learning and memory. Briefly, paraffin-embedded brain tissue sections from control and TAA rats (10 microns), were incubated with anti-ionized calcium-binding adapter molecule 1 (Iba1) antibody (FL-147: sc-98468, 1:200 dilution, Santa Cruz Biotechnology, Inc., Dallas, TX, USA). Sections were washed and incubated and were used at a concentration of 1:200. Immunofluorescent images were acquired with a Zeiss LSM510/UV Axiovert 200M confocal microscope (Carl Zeiss Microscopy, LLC, Thornwood, NY, USA) with a Plan Apochromat 40× objective lens and 2× zoom, resulting in images that were125 μm × 125 μm in area with a 1.0-μm optical slice thickness (1.0 Airy units for Alexa Fluor 546 or 568 emission channel). A random collection of images from sections of control and TAA-treated rats was obtained by systematically capturing each image in a “blind” manner by moving the microscope stage approximately 5 mm in four different directions. At least 14 fluorescent images were captured per rat. Images were quantified using the Velocity 6.0 High-Performance Cellular Imaging Software (PerkinElmer, Waltham, MA, USA), as described previously [35,36,37], sand were normalized to the number of DAPI-positive cells, as well as to the area and intensity of DAPI.

## 3. Statistical Analysis

The statistical analysis was performed using Sigma Stat Software (3.0 version, DataMost Corporation, Chatsworth, CA, USA). Results were expressed as mean ± standard error (SEM). All data were subjected to one-way analysis of variance (ANOVA). Values with *p* < 0.05 were considered as statistically significant.

## 4. Results

### 4.1. Assessment of Liver Function

#### 4.1.1. Liver Morphometry

In the rats with chronic liver failure, we noted a significant increase in the LBWR (*p* < 0.001, Figure 1), compared to the control group, indicating the presence of hepatomegaly.

#### 4.1.2. Biochemical Analysis

In addition to the study of liver morphometry, we carried out a study of liver function markers (AST, ALT). Our results showed a highly significant elevation of ALT (*p* < 0.01, Figure 2A) and AST (*p* < 0.01, Figure 2B) in the TAA-treated group compared to the control. Otherwise, blood urea levels showed a slight increase (Figure 2C) that was not statistically significant. Additionally, we assessed serum creatinine levels as a kidney function marker for revealing eventual renal dysfunction, which is typically associated with advanced liver failure (hepatorenal syndrome). Our results showed a significant decrease in creatinine levels (*p* < 0.05, Figure 2D), which might reflect renal failure [38]. Moreover, we found a significant increase in brain and blood ammonia in TAA-treated rats compared to the control (*p* < 0.05, Figure 3). In addition, alterationsin other serum and liver markers in TAA rats were found (Table 1).

#### 4.1.3. Histopathological Examination

The histopathological examination of the liver tissue in the TAA rats stained with H and E shows the presence of sever tissue damages including centrilobular hepatocyte necrosis, ballooning degeneration, hydropic changes, and the presence of eosinophilic bodies, affecting approximately 60–70% of the liver parenchyma, characteristic of liver cirrhosis and fibrosis (Figure 4).

### 4.2. Measurement of Working Memory Impairment 

#### 4.2.1. Memory Impairment as Measured by Novel Object Recognition Test

Our results showed a reduced time in exploring the novel object (Figure 5A) with a highly significant decrease in the discrimination ratio (*p* < 0.001) in TAA-treated rats, as compared to the controls, as an indication of memory impairment (Figure 5B).

#### 4.2.2. Memory Impairment as Measured by Y-Maze Test

The percentage of alternation behavior in the Y-maze was significantly reduced (*p* < 0.05, Figure 6) in TAA-treated rats, as compared to the controls.

#### 4.2.3. Memory Impairment as Measured by T-Maze Test

The percentage of alternation in the T-maze test was significantly decreased (*p* < 0.001, Figure 7) in the TAA group compared to the controls. 

### 4.3. Immunofluorescence Study

Our results showed a significant increase in Iba1 immunoreactivity, as well as number of cells (equally) in the hippocampal subfields CA1 (*p* < 0.01, Figure 8), CA3 (*p* < 0.001, Figure 9) and DG (*p* < 0.001, Figure 10) in TAA-treated rats, respectively, compared to the controls. 

## 5. Discussion

Our findings demonstrate impairment of short working memory and microglial activation in the hippocampal subfields of type C HE rats. Liver dysfunction was demonstrated in the histopathology and liver morphometric study, by the presence of increased key liver enzymes ALT and AST, as well as by the altered serum and liver biomarkers, which were identical to those observed in cirrhotic livers, in TAA-treated rats.

The levels of liver enzymes ALT and AST were significantly elevated, and the liver morphometry analysis showed an increase in LBWR, indicating an enlargement in liver size (hepatomegaly) in TAA-treated rats compared to the controls. This enlargement of the liver in our TAA-treated rats was previously reported in several studies of HE models induced by TAA administration [22].

Previous reports of this animal model have described: hepatic damage (increased blood levels of alanine aminotransferase and aspartate aminotransferase), histopathological changes in the liver, alterations in brain biochemical parameters (increase in neuronal nitric oxide synthase, NADPH oxidase activity, over-activation of glucose-6-phosphate dehydrogenase, and a decline in phosphofructokinase-2), and behavioral abnormalities characteristic of Type C HE [36,39,40,41]. We further characterized this model and found liver and brain morphological changes (i.e., liver sections from TAA-treated rats showed ballooning degeneration, hydropic changes, and the presence of eosinophilic bodies affecting approximately 60–70% of the liver parenchyma, predominantly in periportal regions). We also found increased blood and brain ammonia levels (two- to three-fold). Ammonia levels in the blood and brain tissue of these rats were comparable with those found in rats after portocaval anastomosis [42,43,44].

Furthermore, we identified altered serum and liver biomarkers in TAA-treated rats identical to those observed in cirrhotic livers. These included serum levels of Apo A1, α-2-Macroglobulin, hyaluronic acid, procollagen III amino terminal peptide, tissue inhibitor of metalloproteinase 1, matrix metalloproteinases 3 and 9, chondrex (YKL40), connective tissue growth factor (CTGF), paraoxonase-1, multidrug resistance protein-2, along with an elevation in aspartate aminotransferase and alanine aminotransferase. In addition to these biomarkers, an altered level of transforming growth factor beta-1 was observed in the livers of TAA-treated rats. These observations provide strong support for the initiation and subsequent development of possible liver fibrosis in this animal model. These findings demonstrate that a low dose of TAA administered for 10 consecutive days reproduces many of the features associated with Type C HE in humans, consistent with observations by other investigators.

The abnormality could be the result of liver cirrhosis, as this is a characteristic feature of this animal model [22]. Thus, cirrhosis can result from the accumulation of extracellular matrix proteins (their production exceeds their elimination) in liver tissue [45]. Likewise, renal failure is among the common complications of cirrhosis, and is known as hepatorenal syndrome. This could explain our results regarding the decreased creatinine level and the slight increase in the urea level (plasma urea clearance) in the TAA-treated group, which might indicate renal failure [38].

Type C HE triggers an obvious decline in short-term working memory, as demonstrated by the loss of alternation behavior in T-maze and Y-maze tests and the decreased exploratory index in the novel object recognition test. Indeed, it is highly established that memory impairment generally occurs in HE, especially in the overt form, and experiential data in different animal models of HE appears to corroborate such an identification. Indeed, alteration in spatial working memory was shown in a type C HE model induced by oral TAA administration in rats [46]. Moreover, rats with type C HE induced by TAA administration (i.p injections of 250 mg/kg TAA for 2 days at 24 h intervals) had difficulty in readily acquiring new information, leading to impaired cognitive flexibility 60 days after the last injection [47]. In another study by Méndez et al. (2008) using a rat model of chronic HE induced by TAA administration, impairment of spatial reference memory as accessed by the Morris water maze (MWM) test [46] was reported. More recently, Zhang and coworkers also detected memory deficits in a rat model of mild HE (MHE) using the MWM test [48].

A significant decrease in the discrimination index using the novel object recognition test, as well as a loss of precise alternations was reported in bile-duct-ligated (BDL) rats using the Y-maze test, suggesting an obvious impaired short working memory [49,50,51]. Further, the involvement of ammonia in the worsening of memory deficits was reported in the Acute-on-Chronic type of HE. Indeed, a decrease in the number of attempts made to avoid foot shock in the active avoidance test was noted in cirrhotic BDL rats with a hyperammonemic diet (HD) (BDL + HD) [52]. Additionally, memory impairment to an aversive stimulus in a passive avoidance task with no intrinsic hepatocellular disease was noted in rats with portacaval shunt (PCS) [53]. A decreased ability to learn a conditional discrimination task in the Y-maze test was observed [54,55], and disrupted spatial memory in MWM was noted [56].

Our immunofluorescence study in the hippocampal subfields of CA1, CA3 and DG showed high expressions of microglia activation in all three hippocampal regions in TAA-treated rats. A review of the literature shows a lack of studies on microglia activation in these brain hippocampal regions with regard to short working memory in chronic HE. For instance, microglia activation was reported in patients and animal models of chronic HE, such as in the BDL or portacaval shunt model [55,57], and even in hyperammonemia diet models without liver injury [17,51,58]. Indeed, in the absence of liver failure, hyperammonemia is thought to be sufficient to elicit central associated peripheral inflammations with microglial activation, which were then reduced by peripheral administration of the anti-inflammatory drug ibuprofen [17]. Otherwise, the specific correlation between the presence of neuroinflammation and HE was established in human patients showing the upregulation of Iba1 in the postmortem cerebral cortex of cirrhotic patients with HE but not in cirrhotic patients without HE [57]. 

It is believed that neuroinflammation might have a crucial role in the development of cognitive impairment in both covert and overt HE arising from chronic liver failure [19]. Indeed, CLD induces neuroinflammation that can contribute to cognitive impairment in HE [19]. Several studies using animal models of MHE have indicated that neuroinflammation is responsible for cognitive impairment [4,59,60,61]. For instance, rats with BDL or PCS exhibited neuroinflammation in the brain and especially the cerebellum, which was associated with cognitive deficits in these animals [17,54]. Furthermore, PCS rats showed microglia activation and reduced ability to learn a Y-maze conditional discrimination task [55]. A recent study by Zhang et al. (2021) demonstrated cognitive impairment with microglia activation (Iba1 in the prefrontal cortex) and the subsequent expression of pro-inflammatory factors (IL-1β, TNF-α, IL-18) in an MHE rat model induced by TAA administration [48]. These proinflammatory cytokines are thought to contribute to cognitive impairment [62,63], and can impair synaptic plasticity in the DG and CA regions of the hippocampus [64,65,66,67,68,69] in addition to disrupting hippocampal-dependent learning and memory [70,71,72]. In one study by Chen et al. (2014), using the BDL model and BDL followed by a diet containing ammonium acetate (BDL + HA), rats displayed spatial learning memory deficits and showed a significant decrease in the density of dendritic spines of CA1 pyramidal neurons with increased activated microglia frequency [51] which could explain our finding regarding short-term working memory impairment. This finding is supported by other studies of HE models. In the type C HE model induced by carbon tetrachloride (CCl4), microglia activation and increased pro-inflammatory cytokines in the CA1 region of the hippocampus were noted, along with spatial learning and memory impairments [73]. More importantly, chronic hyperammonemia was shown to induce neuroinflammation in the hippocampus [4]. Rifaximin treatment used to prevent or even reverse neuroinflammation in the hippocampus appeared to enhance spatial learning and memory [4,59,60,74,75]. This compelling evidence supports the pivotal role of neuroinflammation in cognitive impairment in HE.s

## 6. Conclusions

Through the present study we demonstrated microglia activation within hippocampal subfields CA1, CA3 and DG, which was associated with impaired working memory function in a type C HE rat model induced by the repeated administration of TAA. The memory impairment manifested in our rats appears to have been the result ofa possible microglia activation and subsequent neuroinflammation in the hippocampus. However, the exact molecular patho-mechanism by which it exerts these effects is not fully elucidated and needs further research. 

## Figures and Tables

**Figure 1 metabolites-14-00193-f001:**
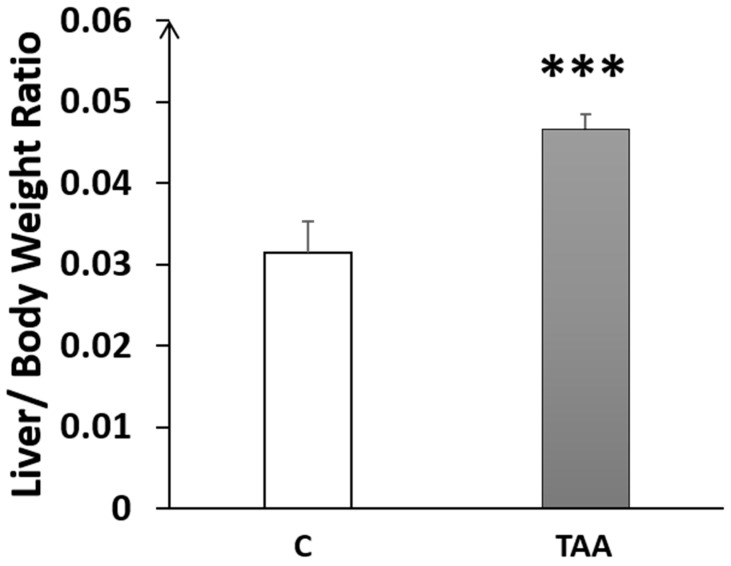
Histogram showing the liver-to-bodyweight ratio (LBWR) in the control (C) and TAA groups. *** *p* < 0.001 vs. control.

**Figure 2 metabolites-14-00193-f002:**
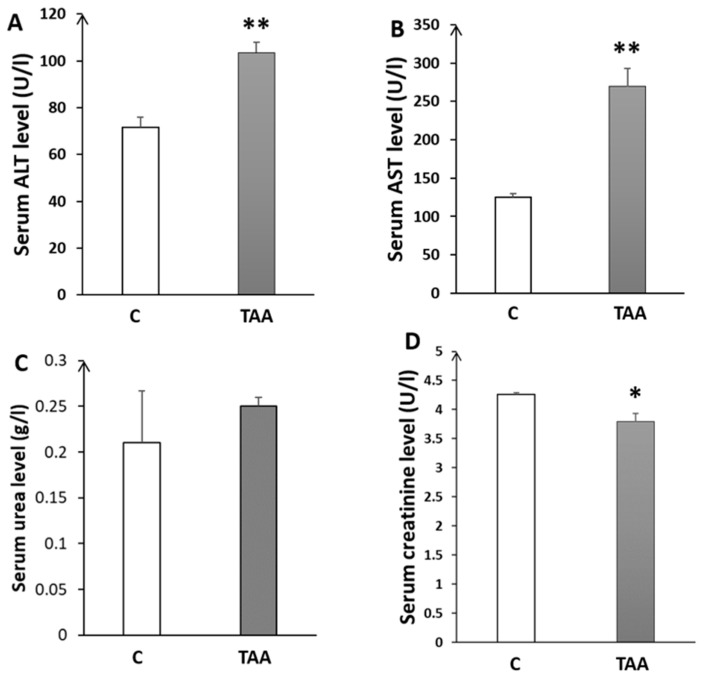
Histograms showing biochemical serum levels in the control (C) and TAA groups: (**A**) alanine aminotransferase; (**B**) aspartate-aminotransferase; (**C**) urea; and (**D**) creatinine. * *p* < 0.05, ** *p* < 0.01 TAA vs. C.

**Figure 3 metabolites-14-00193-f003:**
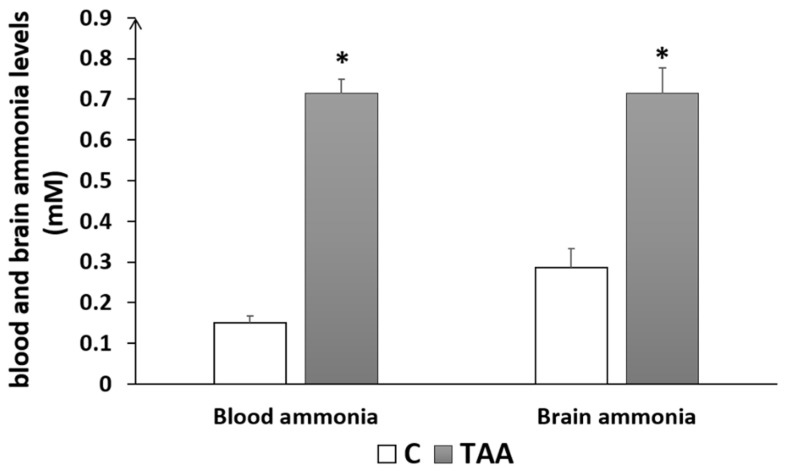
Histogram showing blood and brain ammonia levels of rats with Type C HE compared to controls. * *p* < 0.05 TAA vs. C.

**Figure 4 metabolites-14-00193-f004:**
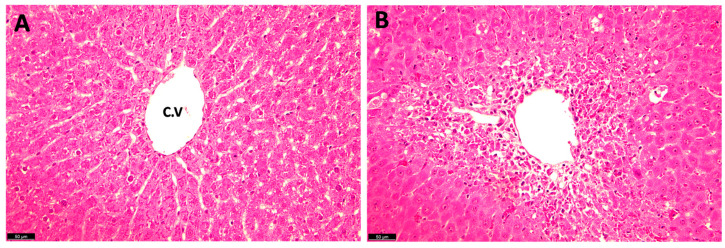
Hematoxylin and eosin (H and E) staining of rat liver sections following TAA administration: (**A**) normal control liver; and (**B**) TAA-treated (10 days) rat liver. C.V: central vein.

**Figure 5 metabolites-14-00193-f005:**
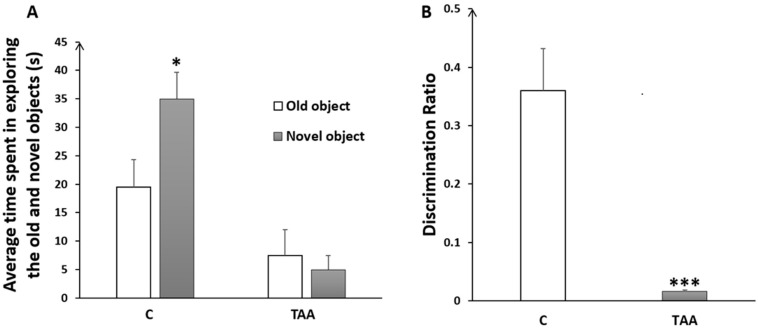
Histogram showing the average time spent in exploring the old and the novel objects (**A**); and the discrimination ratio (**B**) in the control (C) and TAA groups., * *p* < 0.05, *** *p* < 0.001 vs. control.

**Figure 6 metabolites-14-00193-f006:**
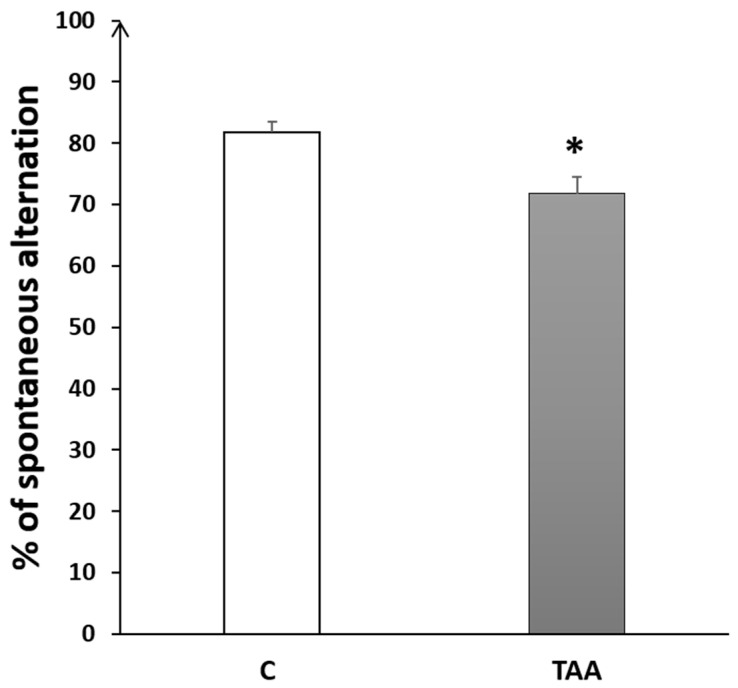
Histogram showing the percentage of alternation behavior in the Y-Maze test in TAA in the control (C) and TAA groups, * *p* < 0.05 vs. control.

**Figure 7 metabolites-14-00193-f007:**
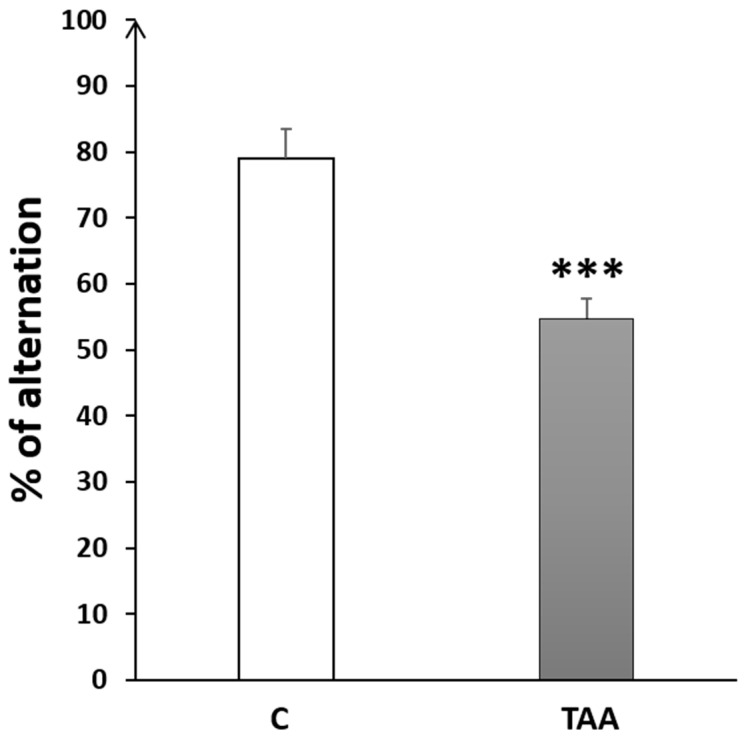
Histogram showing the percentage of alternation behavior in T-Maze task in the control (C) and TAA groups.*** *p* < 0.001 vs. control.

**Figure 8 metabolites-14-00193-f008:**
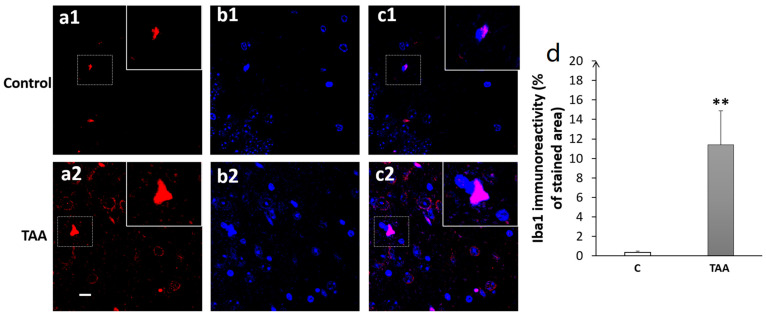
Photomicrographs of the CA1 hippocampal slides immuno-stained with anti-Iba1 (**a1**,**a2**) or anti-DAPI (**b1**,**b2**) and the merged (**c1**,**c2**). (**a1**–**c1**) (controls), (**a2**–**c2**) (TAA). (**d**): Iba1 immunoreactivity (% of stained area as well as number of cells). Scale bar = 35 µm. ** *p* < 0.01 vs. control.

**Figure 9 metabolites-14-00193-f009:**
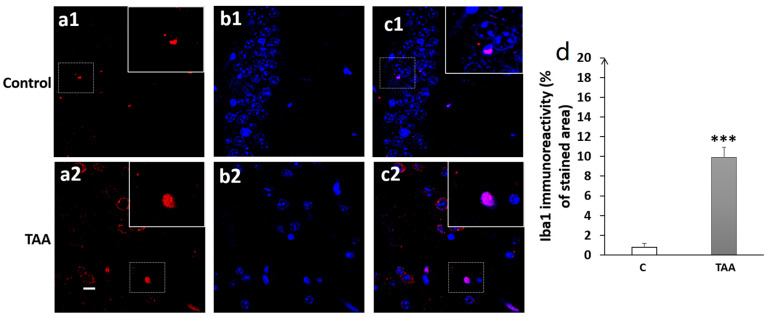
Photomicrographs of the CA3 hippocampal slides immuno-stained with anti-Iba1 (**a1**,**a2**); anti-DAPI (**b1**,**b2**); and merged (**c1**,**c2**). (**a1**–**c1**) (controls), (**a2**–**c2**) (TAA). (**d**) Iba1 immunoreactivity (% of stained area as well as number of cells). Scale bar = 35 µm. *** *p* < 0.001 vs. control.

**Figure 10 metabolites-14-00193-f010:**
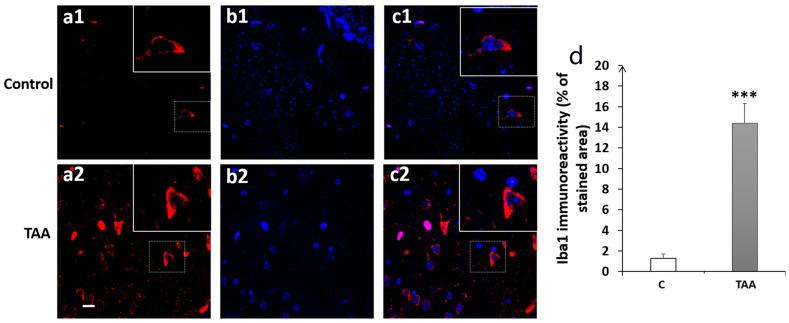
Photomicrographs of the DG hippocampal slides immuno-stained with anti-Iba1 (**a1**,**a2**); anti-DAP (**b1**,**b2**); and merged (**c1**,**c2**). (**a1**–**c1**) (controls), (**a2**–**c2**) (TAA). (**d**) Iba1 immunoreactivity (% of stained area and number of cells). Scale bar = 35 µm. *** *p* < 0.001 vs. control.

**Table 1 metabolites-14-00193-t001:** Serum and liver biomarkers of rats treated with TAA and controls.

	Control	TAA
ApoA1	4.2 ± 0.4 µg/mL	0.25 ± 0.06 µg/mL ***
α-2-Macroglobulin	14.2 ± 2.3 ng/mL	56.3 ± 5.9 ng/mL ***
Hyaluronicacid	41.8 ± 5.9 ng/mL	142.2 ± 13.8 ng/mL ***
Procollagen III amino terminal peptide	17.8 ± 4.1 ng/mL	28.1 ± 2.6 ng/mL *
Tissue inhibitor of metalloproteinase 1	0.56 ± 0.08 ng/mL	1.8 ± 0.2 ng/mL ***
Matrix metalloproteinase-3	162.7 ± 16.7 ng/mL	96.4 ± 12.8 ng/mL **
Matrix metalloproteinase-9	52.3 ± 6.9 ng/mL	36.4 ± 2.1 ng/mL *
YKL40	1.2 ± 0.2 ng/mL	2.7 ± 0.7 ng/mL *
CTGF	98.2 ± 11.6 ng/mL	178.5 ± 36.9 ng/mL **
Paraoxonase-1	3.8 ± 0.5 µg/mL	1.8 ± 0.4 µg/mL ***
TGF BETA	4.1 ± 1.2 pg/mg protein	11.2 ± 3.1 pg/mg protein **

* *p* < 0.05, ** *p* < 0.01, *** *p* < 0.001 TAA vs. Control.

## Data Availability

Data is contained within the article.

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
