# Peer review of "Short Working Memory Impairment Associated with Hippocampal Microglia Activation in Chronic Hepatic Encephalopathy"

_metabolites, 2024, doi:10.3390/metabo14040193_

Round 1
Reviewer 1 Report
Comments and Suggestions for Authors
The study by El-Mansoury et al. focus on hepatic encephalopathy (HE), a condition resulting from liver dysfunction, particularly common in advanced liver disease and cirrhosis patients. The study explores the memory impairment aspect of HE, the underlying mechanisms of which are not yet fully understood, but potentially involve neuroinflammation. Using a chronic HE model in male Wistar rats induced by thioacetamide administration, the research assessed short working memory (SWM) and examined microglial changes. The results revealed impaired SWM in the TAA-treated rats and observed microglial activation in all three hippocampal regions (CA1, CA3, and dentate gyrus), suggesting a potential link between microglial activation and cognitive impairment in HE. This suggests that microglial activation and subsequent neuroinflammation may contribute to memory decline in HE patients. Some revisions are suggested to improve the quality of the manuscript.
1. The model used in the study lacks proper validation. The author opted for a type C hepatic encephalopathy (HE) model induced by TAA, but it is essential to substantiate that this model accurately represents the characteristics of type C HE. Simply referencing protocols from other studies does not guarantee the model's reproducibility in the author's specific research environment. Providing concrete evidence of its suitability for the intended purpose and validation within the author's laboratory would strengthen the study's credibility.
2. TAA is commonly employed to induce fibrosis in animal models. Besides assessing transaminase levels, it is advisable for the author to include histological evidence to confirm liver injury. This can be achieved through techniques such as hematoxylin and eosin (H&E) staining, as well as the evaluation of fibrosis markers like Sirius red staining or immunohistochemistry (IHC) for collagen I. Incorporating these additional analyses would provide a more comprehensive and robust assessment of liver damage in the study.
3. To establish a link between memory impairment and hepatic encephalopathy (HE), it is recommended that the author include measurements of blood ammonia levels. This can help provide valuable insights into the potential relationship between elevated ammonia levels and cognitive deficits in HE animals.
4. Given the potential consequences of microglia activation, it is advisable to include measurements of proinflammatory cytokines. This would enhance the understanding of the inflammatory response associated with microglia activation and its impact on cognitive function in the context of the study.
5. Certain expressions in the manuscript require revision. For instance, in line 241 of the discussion section, the use of "chronic liver failure" may not be accurate, as the transaminase levels do not align with this diagnostic term.
6. Given the limited brain tissue size, accurately identifying the specific locations of hippocampal subfields, including CA1, CA3, and DG, can be challenging. To ensure the precise localization of these regions, it is advisable for the author to provide histological evidence using specific markers to stain the tissue. This would help demonstrate that the designated areas have been correctly identified and analyzed.
Comments on the Quality of English Languagemoderate editing.
Author Response
- The model used in the study lacks proper validation. The author opted for a type C hepatic encephalopathy (HE) model induced by TAA, but it is essential to substantiate that this model accurately represents the characteristics of type C HE. Simply referencing protocols from other studies does not guarantee the model's reproducibility in the author's specific research environment. Providing concrete evidence of its suitability for the intended purpose and validation within the author's laboratory would strengthen the study's credibility.
We agree with the reviewer. We made extensive investigation to validate the model to represent Type C HE. Precisely, we examined whether genes known to modulated in livers of cirrhotic patients also occurring in this model. Accordingly, we performed a comprehensive analyses of liver mRNA expression. We indeed identified many genes that are known to be altered in livers of cirrhotic patients. These include, Apolipoprotein A1 (Apo A1), α-2-Macroglobulin, hyaluronic acid, procollagen III amino terminal peptide, tissue inhibitor of metalloproteinase 1, matrix metalloproteinases 3 (MMP-3) and 9, chondrex (YKL40), connective tissue growth factor (CTGF), paraoxonase-1, multidrug resistance protein-2, transforming growth factor beta-1. We now provide this data in the revised m/s.
- TAA is commonly employed to induce fibrosis in animal models. Besides assessing transaminase levels, it is advisable for the author to include histological evidence to confirm liver injury. This can be achieved through techniques such as hematoxylin and eosin (H&E) staining, as well as the evaluation of fibrosis markers like Sirius red staining or immunohistochemistry (IHC) for collagen I. Incorporating these additional analyses would provide a more comprehensive and robust assessment of liver damage in the study.
We now provide a detailed hematoxylin and eosin (H&E) stained livers from TAA-treated rats in the revised m/s (see Fig. 4).
- To establish a link between memory impairment and hepatic encephalopathy (HE), it is recommended that the author include measurements of blood ammonia levels. This can help provide valuable insights into the potential relationship between elevated ammonia levels and cognitive deficits in HE animals.
This has been now added in the revised m/s (see Fig.3).
- Given the potential consequences of microglia activation, it is advisable to include measurements of proinflammatory cytokines. This would enhance the understanding of the inflammatory response associated with microglia activation and its impact on cognitive function in the context of the study.
While we found increased Iba-1 we failed to consistently identify increase in cytokine levels in (the mRNA level measured in brain of TAA treated rats, IL-1beta, IL-6 and TNF-alpha were inconsistent). Noteworthy, similar findings were observed in ammonia-treated rats as well as in patients with hepatic encephalopathy (Zemtsova I, Görg B, Keitel V, Bidmon HJ, Schrör K, Häussinger D. Microglia activation in hepatic encephalopathy in rats and humans. Hepatology. 2011 Jul;54(1):204-15. doi: 10.1002/hep.24326. PMID: 21452284.). These findings strongly suggest that the significance of microglial activation may likely due to an increased oxidative stress in Type C HE, although increased cytokine levels are consistently identified in acute HE (Type A HE) (Jiang W, Desjardins P, Butterworth RF. Cerebral inflammation contributes to encephalopathy and brain edema in acute liver failure: protective effect of minocycline. J Neurochem. 2009 Apr;109(2):485-93. doi: 10.1111/j.1471-4159.2009.05981.x. Epub 2009 Feb 7. PMID: 19220703.). We now added a note in the revised m/s.
- Certain expressions in the manuscript require revision. For instance, in line 241 of the discussion section, the use of "chronic liver failure" may not be accurate, as the transaminase levels do not align with this diagnostic term.
We have now modified this as “Type C HE”.
- Given the limited brain tissue size, accurately identifying the specific locations of hippocampal subfields, including CA1, CA3, and DG, can be challenging. To ensure the precise localization of these regions, it is advisable for the author to provide histological evidence using specific markers to stain the tissue. This would help demonstrate that the designated areas have been correctly identified and analyzed.
The images were taken by a world-renowned neuroscientist (Dr. Jayakumar, one of the contributing authors in the current m/s) who has extensive experience in identifying brain regions not only in mice and rats, but also in primates and humans. Further, while pictures taken by lower magnification show specific regions, the images are dotted, and it is difficult to identify the specific staining.
Reviewer 2 Report
Comments and Suggestions for Authors
Bilal El-Mansoury et al., is having many english grammatical errors, Please correct them all.
I would suggest authors perform some more experiments such as RTPCR or ELISA for a few more neuroinflammatory markers to conclude neuroinflammation by Ila1 is not enough.
Authors should present all graphs with dotted bar graphs.
Comments on the Quality of English LanguageNeed to be improved
Author Response
- Bilal El-Mansoury et al., is having many english grammatical errors, Please correct them all.
We have performed a thorough language editing.
- I would suggest authors perform some more experiments such as RTPCR or ELISA for a few more neuroinflammatory markers to conclude neuroinflammation by Ila1 is not enough.
While we found increased Iba-1 we failed to consistently identify increase in cytokine levels in (the mRNA level measured in brain of TAA treated rats, IL-1beta, IL-6 and TNF-alpha were inconsistent). Noteworthy, similar findings were observed in ammonia-treated rats as well as in patients with hepatic encephalopathy (Zemtsova I, Görg B, Keitel V, Bidmon HJ, Schrör K, Häussinger D. Microglia activation in hepatic encephalopathy in rats and humans. Hepatology. 2011 Jul;54(1):204-15. doi: 10.1002/hep.24326. PMID: 21452284.). These findings strongly suggest that the significance of microglial activation may likely due to an increased oxidative stress in Type C HE, although increased cytokine levels are consistently identified in acute HE (Type A HE) (Jiang W, Desjardins P, Butterworth RF. Cerebral inflammation contributes to encephalopathy and brain edema in acute liver failure: protective effect of minocycline. J Neurochem. 2009 Apr;109(2):485-93. doi: 10.1111/j.1471-4159.2009.05981.x. Epub 2009 Feb 7. PMID: 19220703.). We now added a note in the revised m/s.
- Authors should present all graphs with dotted bar graphs.
We have now corrected the issue.
Reviewer 3 Report
Comments and Suggestions for Authors
The introduction provides information about hepatic encephalopathy generated by chemical. introduction section need revision.
1. Only the initial look of an abbreviations needs to be defined.
2. Introduction section needs revision. It would be appropriate to describe the pathological mechanism of HE as well as how microglia activate in this disease.
3. The materials and methods of analysis presented in the manuscript described properly.
4. The results are presented in detail and appropriate. The results in tables, graphs, Immunofluorescence; all results are processed statistically and justified.
5. Recent references must be cited in the manuscript.
Author Response
The introduction provides information about hepatic encephalopathy generated by chemical. introduction section need revision.
The introduction section is now revised as per the reviewer’s recommendation.
- Only the initial look of an abbreviations needs to be defined.
Thank you, we considered such remark in our m/s
- Introduction section needs revision. It would be appropriate to describe the pathological mechanism of HE as well as how microglia activate in this disease.
We now added them in the text
- Recent references must be cited in the manuscript.
We now added more recent references relevant to m/s.
In closing, we are most grateful to you and the reviewers for the very insightful, detailed, and helpful comments, all of which we have dealt within this letter, text and in figures. We hope that in its present form you will now find the manuscript suitable for publication in the Journal of Metabolites.
Round 2
Reviewer 1 Report
Comments and Suggestions for Authors
My questions have been well-addressed.
Reviewer 2 Report
Comments and Suggestions for Authors
I recommend publication